# Improving the Capacity of Quantum Dense Coding and the Fidelity of Quantum Teleportation by Weak Measurement and Measurement Reversal

**DOI:** 10.3390/e25050736

**Published:** 2023-04-29

**Authors:** Meijiao Wang, Bing Sun, Lianzhen Cao, Yang Yang, Xia Liu, Xinle Wang, Jiaqiang Zhao

**Affiliations:** School of Physics and Electronic Information, Weifang University, Weifang 261061, China

**Keywords:** quantum dense coding, quantum teleportation, capacity, fidelity, weak measurement and measurement reversal

## Abstract

A protective scheme of quantum dense coding and quantum teleportation of the X-type initial state is proposed in amplitude damping noisy channel with memory using weak measurement and measurement reversal. Compared with the noisy channel without memory, the memory factor improves both the capacity of quantum dense coding and the fidelity of the quantum teleportation to a certain extent for the given damping coefficient. Although the memory factor can inhibit decoherence in some degree, it cannot eliminate it completely. In order to further overcome the influence of the damping coefficient, the weak measurement protective scheme is proposed, which found that the capacity and the fidelity can be efficiently improved by adjusting weak measurement parameter. Another practical conclusion is that, among the three initial states, the weak measurement protective scheme has the best protective effect on the Bell-state in terms of the capacity and the fidelity. For the channel with no memory and full memory, the channel capacity of quantum dense coding reaches two and the fidelity of quantum teleportation reaches one for the bit system; the Bell system can recover the initial state completely with a certain probability. It can be seen that the entanglement of the system can be well protected by the weak measurement scheme, which provides a good support for the realization of quantum communication.

## 1. Introduction

Quantum information (QI) science is an emerging field of science and technology formed by the integration of physical science and information science and other disciplines. QI science is a new way of computing, encoding and transmitting information through various coherent properties of quantum system, such as quantum parallelism [1], quantum entanglement [2,3] and quantum noncloning [4]. The research field of QI mainly includes quantum communication [5,6,7], quantum computing [8,9] and quantum precision measurement [10]. The two aspects of quantum communication, which are quantum dense coding and quantum teleportation, are especially discussed here. Quantum entanglement is the core factor of quantum communication, but the system inevitably interacts with its surrounding noisy channel, resulting in the attenuation of quantum entanglement and even the entanglement sudden death (ESD) [11,12,13,14]. Any physical process can be regarded as a quantum noisy channel, which reflects the evolution of the system from the initial state to the final state. So it is an important task for quantum communication to protect entanglement from the influence of the noisy channel.

The physical mechanism of using the weak measurement (WM) protective scheme is that the WM operation causes the system to collapse to the ground state with a certain probability which will not be affected through the amplitude damping (AD) noisy channel, and then the system is restored to the initial state with a certain probability through the quantum measurement reversal (QMR) operation. Although this approach is associated with a lower probability of success, it is very powerful for increasing the entanglement of the system. Since Aharonov et al. [15,16] made the pioneering work, the protective scheme combining weak measurement and measurement reversal (WMR) has been studied extensively in theory and in experiment. For example, the WMR protective scheme have been used to protect the entanglement of one-qubit [17,18,19], two-qubit [20,21,22,23], a hybrid qubit-qutrit [24] and two-qutrit system [25] in quantum noisy channel. The results of the above literature works have proved that the decoherence can be effectively suppressed and the entanglement can be effectively protected by the WMR protective scheme. Even for some states, the entanglement can be completely recovered with a low probability of success. In addition to protecting the entanglement of different quantum systems in noisy channel, quantum dense coding and quantum teleportation are also studied. For example, Tian et al. [26] proposed a protocol of quantum dense coding protection of two qubits based on the WMR protective scheme. The results show that the capacity of quantum dense coding under the WMR protective scheme is always greater than that without the WMR protective scheme. When the protocol is applied, for the AD noisy channels with different damping coefficients, the result reflects that quantum entanglement can be protected and quantum dense coding becomes successful. Li et al. [27] improved the quantum teleportation in the AD noisy channel by the WMR protective scheme, the results indicate that the combination of WM and QMR could drastically enhance the fidelity in AD noisy channel. In addition, Li et al. [28] further explored the fidelity of the system in the Pauli channel, but did not adopt the WMR protective scheme to explore whether the fidelity of the system could be improved. It is worth pointing out that the initial state involved in the refs. [26,27,28] is one of the Bell states.

Through the application of the WMR protective scheme in the above aspects, we can find its broad prospect, and also provide people with a new way to solve the problem that could not be solved by the classical method before. The early researches focus on the assumption that the channel acts as a memoryless configuration, which acts identically and independently on the qubits. However, there are actually two kinds of quantum channels, with memory and without memory. In many realistic scenarios, the Kraus operators of the noisy channel map cannot be expressed in a tenor product form, which means that the noisy channel has memory or is correlated among consecutive uses, so memory or correlated noisy channels appear to be more reasonable and significant in QI theory. Meanwhile, the application of WMR protective scheme in various quantum noisy channels with memory has not been studied. So, we have attempted to study the evolution of the entanglement in four noisy channels with memory and the protection of the entanglement by the WMR protective scheme [29], in which the memory effects are characterized by a memory parameter, which ranges from 0 to 1. The evolution of the entanglement under the correlated channel is investigated with and without the assistance of WMR protective scheme. We derive the exact expressions of the entanglement and find that the WMR protective scheme indeed helps to protect the entanglement from different correlated decoherence channels for two initial Bell-like states. What is more significant is that the WMR protective scheme and improving memory parameters can not only effectively circumvent ESD but also amplify the initial entanglement, which is rather significant in QI science.

Based on the results obtained by other researchers and ourselves, in this paper, we want to expand further and continue to use WMR protective scheme to improve the capacity of quantum dense coding and the fidelity of the quantum teleportation with the X-type initial state which suffers from the AD noisy channels with memory as shown in Figure 1. The memory parameter can improve the capacity and the fidelity to a certain degree. Moreover, after introducing the WMR protective scheme, choosing the appropriate WM strength can further improve the capacity and the fidelity, which works best for the Bell state. For the channel with no memory and full memory, the channel capacity of quantum dense coding reaches 2 and the fidelity of quantum teleportation reaches 1. Our results extend the ability of WM as a technique in various QI processes which are affected by correlated noise.

The rest of this paper is structured as follows. The evolution of the system in the AD correlated noisy channel without and with WMR operation are introduced in Section 2. In Section 3 is devoted to quantum dense coding and quantum teleportation under the AD noisy channel with memory. In Section 4, the effect of WM strength on the capacity and the fidelity after the introduction of WMR protective scheme. The conclusion is presented in Section 5.

## 2. The Evolution of the System

### 2.1. The Evolution of the System in the AD Noisy Channel with Memory

We assume that the initially entangled state is defined as
(1)$$\rho_{0}=\frac{1}{4}\left(I⊗I+\sum_{i=0}^{3}c_{i}\sigma_{i}^{A}⊗\sigma_{i}^{B}\right),$$
where *I* is the unit operator of two-qubit, $\sigma_{i}^{A}$ and $\sigma_{i}^{B}$ are the Pauli operators for two-qubit of the Alice and Bob. $c_{i}$ ($0⩽c_{i}⩽1$) is the real number that makes the matrix $\rho_{0}$ normalized and positive. In order to study the capacity of quantum dense coding and the fidelity of quantum teleportation, we will consider three cases from the initial state. That is the general state ($|c_{1}|=0.6$, $|c_{2}|=0.7$, $|c_{3}|=0.4$), the Werner state ($|c_{1}\left|=\right|c_{2}\left|=\right|c_{3}|=0.8$), and the Bell state ($|c_{1}\left|=\right|c_{2}\left|=\right|c_{3}|=1$).

In Figure 1 (the solid path labeled ①), the evolution mechanism of the initial entangled state under the AD noisy channel with memory can be expressed as
(2)$$\rho_{AB}=\varepsilon \left(\rho_{0}\right)=\left(1-\eta \right)\sum_{i,j=0}^{1}E_{ij}\rho_{0}E_{ij}^{+}+\eta \sum_{k=0}^{1}A_{k}\rho_{0}A_{k}^{+},$$
where η (0⩽η⩽1) denotes memory effect, $E_{ij}=E_{i}⊗E_{j}$ is the tensor product of the memoryless operator in the AD noisy channel, with
(3)$$\begin{array}{cc} E_{0} & =\left(\begin{array}{cc} 1 & 0\\ 0 & \sqrt{1-r} \end{array}\right),E_{1}=\left(\begin{array}{cc} 0 & \sqrt{r}\\ 0 & 0 \end{array}\right), \end{array}$$
where r∈[0,1] represents the strength of the decoherence. $A_{k}$ is the correlated noise, with the following form
(4)$$\begin{array}{cc} A_{0} & =\left(\begin{array}{cccc} 1 & 0 & 0 & 0\\ 0 & 1 & 0 & 0\\ 0 & 0 & 1 & 0\\ 0 & 0 & 0 & \sqrt{1-r} \end{array}\right),A_{1}=\left(\begin{array}{cccc} 0 & 0 & 0 & \sqrt{r}\\ 0 & 0 & 0 & 0\\ 0 & 0 & 0 & 0\\ 0 & 0 & 0 & 0 \end{array}\right). \end{array}$$

Substituting Equations (3) and (4) into Equation (Equation 2), and the specific form of the evolution process of the entangled system is expressed as
(5)$$\begin{array}{cc} \rho_{AB} & =\frac{1}{4}\left(\begin{array}{cccc} a & 0 & 0 & b\\ 0 & c & d & 0\\ 0 & d^{*} & e & 0\\ b^{*} & 0 & 0 & f \end{array}\right), \end{array}$$
with
(6)$$\begin{array}{cc} a & =\left(1-\eta \right)\left[\left(1+c_{3}\right)\left(1+r^{2}\right)+2\left(1-c_{3}\right)r\right]+\eta \left(1+c_{3}\right)\left(1+r\right),\\ b & =b^{*}=\left(c_{1}-c_{2}\right)\left[\left(1-\eta \right)\left(1-r\right)+\eta \sqrt{1-r}\right],\\ c & =e=\left(1-\eta \right)\left[\left(1-c_{3}\right)\left(1-r\right)+\left(1+c_{3}\right)r\left(1-r\right)\right]+\eta \left(1-c_{3}\right),\\ d & =d^{*}=\left(c_{1}+c_{2}\right)\left[\left(1-\eta \right)\left(1-r\right)+\eta \right],\\ f & =\left(1+c_{3}\right)\left[\left(1-\eta \right){\left(1-r\right)}^{2}+\eta \left(1-r\right)\right]. \end{array}$$

### 2.2. The Evolution of the System after WM and QMR Operation

WM is a kind of local collapse measurement based on von Neumann measurement and positive operator value measurement (POVM). WM satisfies two key requirements, that is, controllability of the measurement intensity and no loss of coherence when the system is slightly disturbed, which is common in any measurement. We know that the WM is limited by the information extracted from the quantum system, which can effectively prevent the quantum state of the measurement system from randomly collapsing into its eigenstate. With WM, we get only partial information about the system, but can keep the system active (i.e., not completely broken). Therefore, the system information can be recovered by some operations of the QMR. In the actual implementation process, the WM is implemented by the detector indirectly monitoring a qubit. If the detector responds, we know that the qubit goes from |1〉 state to |0〉 state; discard this measurement. If the detector does not respond, that means the quantum state has only partially collapsed, and we will leave it to evolve. This unresponsive WM is different from AD measurement, in a sense, it is equivalent to adding an ideal detector to observe changes in the environment.

So, we perform the WMR protective scheme to suppress the entanglement decay. For the qubit case, the WM can be written as a non-unitary quantum operation
(7)$$M_{WM}=\left(\begin{array}{cc} 1 & 0\\ 0 & \sqrt{1-p} \end{array}\right),$$
where p∈[0,1] is the WM strength for the qubit. The QMR for the qubit is also a non-unitary quantum operation that can be written as
(8)$$M_{QMR}=\left(\begin{array}{cc} \sqrt{1-q} & 0\\ 0 & 1 \end{array}\right),$$
where q∈[0,1] is the strength of QMR.

If the system passes through the AD noisy channel with memory directly, the entanglement will decay rapidly. In order to overcome this phenomenon, the WM operation is carried out before the system passes through the AD noisy channel as shown in Figure 1 (the dashed path labeled ②). Then, the QMR operation is implemented, and the final evolution result of the system is
(9)$$\rho_{WMR}=\left(M_{QMR}⊗M_{QMR}\right)\varepsilon [\left(M_{WM}⊗M_{WM}\right)\rho_{0}{\left(M_{WM}⊗M_{WM}\right)}^{+}]{\left(M_{QMR}⊗M_{QMR}\right)}^{+},$$

By combining Equations (1) and (7)–(9), the density matrix of the system can be obtained as
(10)$$\begin{array}{cc} \rho_{WMR} & =\frac{1}{P}\left(\begin{array}{cccc} a^{{}^{\prime }} & 0 & 0 & b^{{}^{\prime }}\\ 0 & c^{{}^{\prime }} & d^{{}^{\prime }} & 0\\ 0 & d^{{*}^{\prime }} & e^{{}^{\prime }} & 0\\ b^{{*}^{\prime }} & 0 & 0 & f^{{}^{\prime }} \end{array}\right), \end{array}$$
with
(11)$$\begin{array}{cc} a^{{}^{\prime }} & =\frac{1}{4}\left\{\left(1-\eta \right)\left[\left(1+c_{3}\right)+\left(1+c_{3}\right)r^{2}{\left(1-p\right)}^{2}+2\left(1-c_{3}\right)r\left(1-p\right)\right]+\eta \left[\left(1+c_{3}\right)+\left(1+c_{3}\right)r{\left(1-p\right)}^{2}\right]\right\}{\left(1-q\right)}^{2},\\ b^{{}^{\prime }} & =b^{{*}^{\prime }}=\frac{c_{1}-c_{2}}{4}\left[\left(1-\eta \right)\left(1-r\right)\left(1-p\right)+\eta \sqrt{1-r}\left(1-p\right)\right]\left(1-q\right),\\ c^{{}^{\prime }} & =e^{{}^{\prime }}=\frac{1}{4}\left\{\left(1-\eta \right)\left[\left(1-c_{3}\right)\left(1-r\right)\left(1-p\right)+\left(1+c_{3}\right)r\left(1-r\right){\left(1-p\right)}^{2}\right]+\eta \left(1-c_{3}\right)\left(1-p\right)\right\}\left(1-q\right),\\ d^{{}^{\prime }} & =d^{{*}^{\prime }}=\frac{c_{1}+c_{2}}{4}\left[\left(1-\eta \right)\left(1-r\right)\left(1-p\right)+\eta \left(1-p\right)\right]\left(1-q\right),\\ f^{{}^{\prime }} & =\frac{1+c_{3}}{4}\left[\left(1-\eta \right){\left(1-r\right)}^{2}+\eta \left(1-r\right)\right]{\left(1-p\right)}^{2},\\ P & =a^{{}^{\prime }}+c^{{}^{\prime }}+e^{{}^{\prime }}+f^{{}^{\prime }}. \end{array}$$

## 3. Quantum Dense Coding and Quantum Teleportation under the AD Noisy Channel with Memory

### 3.1. Quantum Dense Coding

Quantum dense coding is one of the most important quantum secure communication processes. Take the bit system as an example. At the beginning, Alice and Bob share a pair of entangled photons. Alice encodes 2-bits of classical information on its photon and sends the photon to Bob, who then performs Bell-state measurement on the two photons in his hand and decodes the 2-bits of information sent by Alice. This process not only transmits classical information but also increases the source coding capacity exponentially through quantum dense coding. The idea of quantum dense coding was proposed in 1992 and first realized in optical systems in 1996 [30]. An important measure of quantum dense coding is the channel capacity, the capacity of quantum dense coding can be measured by the Holevo quantity
(12)$$\chi =S\left(\rho^{*}\right)-S\left(\rho \right),$$
here ρ represents the density matrix of the two-qubit A and B, $\rho^{*}$ denotes the density matrix of the system after quantum dense coding. χ stands for the capacity of quantum dense coding and *S* is the von Neumann entropy, which can be expressed as
(13)$$S\left(\rho \right)=-\sum_{i}\lambda_{i}log_{2}\lambda_{i},$$
where $\lambda_{i}$ are the eigenvalues of the density matrix ρ.

If the basis vector of the orthogonal unitary transformation is {|0〉,|1〉}, the dense coding process of two qubits is expressed as
(14)$$\begin{array}{cc} U_{00}|x\rangle & =|x\rangle ,\\ U_{10}|x\rangle & =e^{i\pi x}|x\rangle ,\\ U_{01}|x\rangle & =|x+1(mod2)\rangle ,\\ U_{11}|x\rangle & =e^{i\pi x}|x+1\left(mod2\right)\rangle . \end{array}$$

In combination with Equation (Equation 14), after dense coding, the density matrix $\rho^{*}$ of the system is [31,32]
(15)$$\rho^{*}=\frac{1}{4}\sum_{0}^{3}\left(U_{i}⊗I\right)\rho_{0}\left(U_{i}^{+}⊗I\right),$$
supposing 0→00; 1→01; 2→10; 3→11.

As shown in Figure 1a (the solid path labeled ①), the capacity $\chi_{1}^{\psi }$ of quantum dense coding for the density matrix $\rho_{AB}$ in Equation (Equation 5) is
(16)$$\begin{array}{cc} \chi_{1}^{\psi } & =S\left(\rho_{AB}^{*}\right)-S\left(\rho_{AB}\right)\\ \; & \;=-\left(a+e\right)log_{2}\left(\frac{a+e}{2}\right)-\left(c+f\right)log_{2}\left(\frac{c+f}{2}\right)+\sum_{i=1}^{4}\lambda_{i}log_{2}\lambda_{i}, \end{array}$$
where
(17)$$\begin{array}{cc} \lambda_{1,2} & =\frac{1}{2}\left(c+e\pm \sqrt{c^{2}+4d^{2}+e^{2}-2ce}\right),\\ \lambda_{3,4} & =\frac{1}{2}\left(a+f\pm \sqrt{a^{2}+4b^{2}+f^{2}-2af}\right). \end{array}$$

The capacity $\chi_{1}^{\psi }$ of quantum dense coding as the function of the damping coefficient *r* and the memory parameter η for different initial states, such as Bell state, Werner state, and General state, are plotted in Figure 2a. It can be seen from Figure 2a that, for the three different initial states, the capacity $\chi_{1}^{\psi }$ decreases first and then increases as damping coefficient *r* increases, but if the channel takes the same parameter, the capacity $\chi_{1}^{\psi }$ of Bell state is the strongest.

In order to show the change of the capacity $\chi_{1}^{\psi }$ more intuitively, the capacity $\chi_{1}^{\psi }$ with damping coefficient *r* in three initial states with different memory coefficients η is shown in Figure 3. From Figure 3a, $\chi_{1}^{B\psi }$ firstly decreases and then increases with the increase of *r*, eventually which maintains a constant value $\chi_{1}^{B\psi }=1$ at r=1 when 0≤η<1. For η=1, that is fully memory amplitude damping channel, $\chi_{1}^{B\psi }$ decreases with *r* increasing until $\chi_{1}^{B\psi }=1$. In addition, $\chi_{1}^{B\psi }$ increases with η for a fixed *r*. In Figure 3b,c, the capacity $\chi_{1}^{W(G)\psi }$ of Werner and General states is obviously weaker than that of Bell state. In Figure 3b, the capacity $\chi_{1}^{W\psi }$ decreases with the increase in memory coefficient η with r≥0.9. In Figure 3c, the capacity $\chi_{1}^{G\psi }$ decreases with the increase in memory coefficient η with r≥0.6.

Since $\chi_{1}^{\psi }\leq 1$ means that quantum dense coding is unsuccessful, it can be seen that the capacity $\chi_{1}^{\psi }$ of Bell state is the strongest under the same channel combined with Figure 2 and Figure 3. Although the capacity $\chi_{1}^{\psi }$ increases with the increase of memory coefficient η, the capacity $\chi_{1}^{\psi }$ decreases gradually due to the existence of damping coefficient *r*. Therefore, it is necessary to take further corresponding protective schemes.

### 3.2. Quantum Teleportation

Quantum teleportation is an important communication method for conveying quantum states. It can transmit unknown quantum states to any distance by means of quantum entanglement. Quantum teleportation does not transmit any matter or energy (classical information) but rather the quantum information carried by quantum states. To realize quantum teleportation, the receiver and the sender are first required to have a pair of shared entangled particles. The sender will distinguish the particle of the quantum state to be transmitted (generally not associated with entangled particles) from the entangled particle in its own hand. In this way, the entangled particles on the receiver will suddenly collapse into another state (which state depends on the measurements of the sender). The sender transmits the measurement result to the receiver through the classical channel, and the receiver can recover the original information by performing corresponding unitary transformation on the entangled particle pair it owns.

If the initially entangled state between Alice and Bob is $\rho_{0}$ in Equation (Equation 1) and the quantum state that Alice wants to transmit to Bob is
(18)$$|\psi_{in}\rangle =\cos\frac{\theta }{2}|0\rangle +\sin\frac{\theta }{2}e^{i\phi }|1\rangle .$$ Then, after the AD noisy channel with memory and the procedure of quantum teleportation shown in Figure 1b (the solid path labeled ①), Bob finally gets the output state $\rho_{out}$, which is expressed as
(19)$$\rho_{out}=\left(\begin{array}{cc} \rho_{11} & \rho_{12}\\ \rho_{21} & \rho_{22} \end{array}\right),$$
where
(20)$$\begin{array}{cc} \rho_{11} & =\cos^{2}\frac{\theta }{2}\left(a+f\right)+\sin^{2}\frac{\theta }{2}\left(c+e\right),\\ \rho_{12} & =\sin\frac{\theta }{2}\cos\frac{\theta }{2}e^{-i\phi }\left(b+b^{*}\right)+\sin\frac{\theta }{2}\cos\frac{\theta }{2}e^{i\phi }\left(d+d^{*}\right),\\ \rho_{21} & =\sin\frac{\theta }{2}\cos\frac{\theta }{2}e^{i\phi }\left(b+b^{*}\right)+\sin\frac{\theta }{2}\cos\frac{\theta }{2}e^{-i\phi }\left(d+d^{*}\right),\\ \rho_{22} & =\sin^{2}\frac{\theta }{2}\left(a+f\right)+\cos^{2}\frac{\theta }{2}\left(c+e\right). \end{array}$$

The fidelity is used to measure quantum teleportation, and the fidelity is proportional to quantum entanglement. The greater the entanglement, the greater the fidelity. The fidelity is
(21)$$F^{\psi }=\langle \psi_{in}|\rho_{out}|\psi_{in}\rangle .$$

By substituting Equations (18)–(20) into Equation (Equation 21), the fidelity is expressed as
(22)$$F^{\psi }\left(\theta ,\phi \right)=a+f+\frac{1}{2}\sin^{2}\theta \left[c+e+b+b^{*}-\left(a+f\right)+\cos2\phi \left(d+d^{*}\right)\right].$$ Here, the fidelity $F^{\psi }\left(\theta ,\phi \right)$ depends on the polar θ and the phase parameters ϕ. The transmitted state is unknown, so the fidelity can be averaged as
(23)$$F_{av}^{\psi }=\frac{1}{4\pi }\int_{0}^{\pi }d\theta \int_{0}^{2\pi }d\phi F^{\psi }\left(\theta ,\phi \right)\sin\theta .$$

Taking Equation (Equation 22) into Equation (Equation 23), one can obtain the average fidelity $F_{av}^{\psi }$
(24)$$F_{av}^{\psi }=\frac{1}{3}\left[2\left(a+f\right)+c+e+b+b^{*}\right].$$

The fidelity $F_{av}^{\psi }$ of quantum teleportation as the function of the damping coefficient *r* and the memory parameter η for three initial states is plotted in Figure 2b. It can be seen from Figure 2b that, for the three different initial states, the fidelity $F_{av}^{\psi }$ decreases with the increase of damping coefficient *r* in addition to η=0, but if the channel takes the same parameter, the fidelity $F_{av}^{\psi }$ of Bell state is the optimum.

In order to show the change of the fidelity $F_{av}^{\psi }$ more intuitively, the fidelity $F_{av}^{\psi }$ with damping coefficient *r* in three initial states with different memory coefficients η is shown in Figure 4. As can be seen from Figure 4a, when the initial state is Bell state, the fidelity $F_{av}^{B\psi }$ increases with the enhancement of memory coefficient η. When the initial state is Werner state in Figure 4b, the fidelity $F_{av}^{W\psi }$ decreases with η increasing that *r* is close to 1. For the General state in Figure 4c, when *r* is small, $F_{av}^{G\psi }$ increases with η increasing, while *r* is large, $F_{av}^{G\psi }$ decreases with η increasing.

It can be seen from the results that the memory factor does improve the fidelity of the system if appropriate damping coefficient is selected. However, the decoherence cannot be completely eliminated, so further protective scheme is also needed.

## 4. Quantum Dense Coding and Quantum Teleportation with WMR Protective Scheme under the AD Noisy Channel with Memory

From the above results for the capacity of quantum dense coding and the fidelity of quantum teleportation with the evolution of damping coefficient and memory coefficient, the effect for the initial Bell state is the best. Moreover, the existence of memory factors can effectively inhibit the decoherence of the system. In fact, both quantum dense coding and quantum teleportation are applications of quantum entanglement in QI processing, and when we consider memory factor in the AD noisy channel, it has two effects on the entanglement: one is to suppress the entanglement degradation, the other is to recover the entanglement after a short death. This result is similar to the entanglement dynamics in non-Markov environments, where memory effects also lead to the recovery of the entanglement. Although the memory coefficient can improve the capacity of quantum dense coding and the fidelity of quantum teleportation, the influence brought by AD noisy channel with memory still cannot be eliminated. In order to overcome the negative effect of damping coefficient, we use WMR protective scheme in Equations (7)–(10) to improve the capacity and the fidelity.

### 4.1. Quantum Dense Coding with WMR

Combining Equations (1), (2), (7) and (15), we can get the capacity $\chi_{2}^{\psi }$ with the WMR protective scheme passing the AD noisy channel with memory in Figure 1a (the dashed path labeled ②). In Figure 5a, considering that the capacity $\chi_{2}^{\psi }$ of quantum dense coding for three initial states is the functions of the WM strength *p* and the memory parameter η with r=0.5. Among the three initial states, the Bell state has the best capacity, which increases with the increase of *p* and η.

In Figure 6, the relationship between the capacity $\chi_{2}^{\psi }$ and WM strength *p* under different memory coefficients η is given. As can be seen from Figure 6, the capacity $\chi_{2}^{\psi }$ increases with the enhancement of WM strength *p* for three initial states. When the initial state is Bell state, the capacity $\chi_{2}^{B\psi }$ increases with the enhancement of memory coefficient η, even the channel capacity reaches 2 under both full memory and no memory. When the initial state is Werner state, the capacity $\chi_{2}^{W\psi }$ decreases with η increasing that *p* is greater than 0.9. For the General state, when *p* is less than 0.7, $\chi_{2}^{G\psi }$ increases with η increasing, while *p* takes a large value, $\chi_{2}^{G\psi }$ decreases with η increasing. Although $\chi_{2}^{W(G)\psi }$ decreases with the increase of η when *p* is very high, the capacity $\chi_{2}^{W(G)\psi }$ with WMR protection is higher than that without protection.

### 4.2. Quantum Teleportation with WMR

Combining Equations (1), (2), (7) and (21), we can get the fidelity $F_{av}^{opt\psi }$ with the WMR protective scheme as in Figure 1b (the dashed path labeled ②). In Figure 5b, for the three initial states, the Bell state has the maximum fidelity, which increases with the increase of *p* and η. As can be seen from Figure 7, the fidelity $F_{av}^{opt\psi }$ increase with the enhancement of WM strength *p* for three initial states. When the initial state is Bell state, $F_{av}^{optB\psi }$ increases with the enhancement of η, even the fidelity reaches 1 under both full memory and no memory. When the initial state is Werner state, $F_{av}^{optW\psi }$ decreases with η increasing that *p* is greater than 0.84. For the General state, when *p* is greater than 0.66, $F_{av}^{optG\psi }$ decreases with η increasing. Although the fidelity $F_{av}^{optW(G)\psi }$ decreases with the increase of η when *p* is very high, $F_{av}^{optW(G)\psi }$ with WMR protection is higher than that without protection.

It’s important to point out that for the Bell state, the maximal capacity $\chi_{2}^{B\psi }$ is always equal to 2 and the maximal fidelity $F_{av}^{optB\psi }$ is always equal to 1 when η=0,1 no matter what *r* is. However, for the intermediate case 0<η<1, $\chi_{2}^{B\psi }$ is less than 2 and $F_{av}^{optB\psi }$ is less than 1. Since there is only one dissipative channel in the decoherence process for both memoryless and fully memory AD noisy channels, it ensures that the original state can be restored by QMR. For the general AD noisy channel with memory, there are two dissipative channels in the decoherence process, in which QMR cannot be accurately distinguished so that $\chi_{2}^{B\psi }$ cannot reach 2 and $F_{av}^{optB\psi }$ also cannot reach 1.

## 5. Conclusions

In this paper, we study a method to improve the capacity of quantum dense coding and the fidelity of quantum teleportation in AD noisy channel with memory for X-type initial state. Where different initial states are considered such as Bell state, Werner state and General state. The results show that the memory coefficient is useful for improving both the capacity and the fidelity to a certain extent. However, the decoherence cannot be completely eliminated, so further protective scheme is also needed. Subsequently, we introduce the WMR protective scheme to further improve the capacity and the fidelity. Among the three initial states, the WMR protective scheme has the best protective effect on the Bell state, no matter the capacity or the fidelity. In addition, we can improve the capacity and the fidelity by adjusting damping coefficient, the memory parameter, and the WM strength. By applying the combination of the WM and QMR, the decoherence can be efficiently suppressed. The capacity of quantum dense coding and the fidelity of quantum teleportation, that is the quantum entanglement, can be efficiently improved for the AD noisy channel with memory. Especially for the Bell state, the value of the capacity and the fidelity reaches the limit under both full memory and no memory by the WMR protective scheme. That is, in the bit system, the channel capacity of quantum dense coding reaches 2, and the fidelity of quantum teleportation reaches 1.

Compared with the two-dimensional entanglement of the bit systems, the high-dimensional entanglement has the advantages of high channel capacity and strong resistance to eavesdropping, and has been widely concerned by the academic community in recent years. Ref. [33] improved the channel capacity record of quantum dense coding to 2.09, exceeding the theoretical limit 2 that two-dimensional entanglement can achieve, fully demonstrating the advantages of high-dimensional entanglement in quantum communication, and laying an important foundation for the in-depth study of the high-dimensional entanglement in the field of QI. Based on the success of the WMR protective scheme in improving the fidelity and capacity of the two-dimensional bit system, it is believed that our research will also improve the quantum entanglement of the high-dimensional system. It also lays an important foundation for the realization of quantum teleportation. After all, quantum teleportation experiment is the core functional unit of long-distance quantum communication and distributed quantum computing, as well as the prerequisite research for the feasibility of a global quantum communication network.

In addition to the study of high-dimensional systems, the evolution of the channel with memory is also explored in fields including the scheduling of quantum teleportation [34], the learning of quantum memory [35], the storage of Gaussian multi-antenna system [36], the influence of data freshness [37] and so on. It can be seen that the current research on the channel with memory has certain significance and potential application prospects.

## Figures and Tables

**Figure 1 entropy-25-00736-f001:**
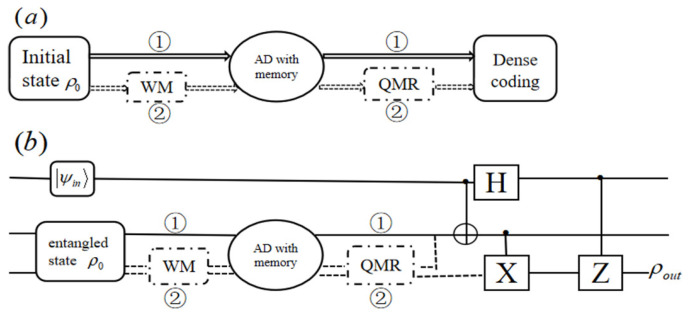
(**a**) The schematic diagram of quantum dense coding; (**b**) The diagram of quantum teleportation. The virtual box represents the weak measurement and measurement reversal operation.

**Figure 2 entropy-25-00736-f002:**
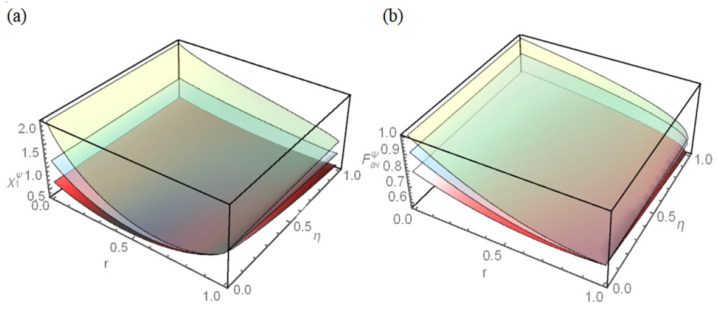
(**a**) The capacity $\chi_{1}^{\psi }$ of quantum dense coding and (**b**) the fidelity $F_{av}^{\psi }$ of quantum teleportation for three initial states as the functions of the damping coefficient *r* and the memory parameter η. The upper layer (green) represents the Bell state, the middle layer (gray) represents the Werner state, and the lower layer (red) represents the General state.

**Figure 3 entropy-25-00736-f003:**
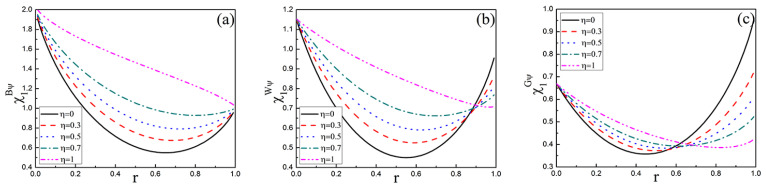
The capacity $\chi_{1}^{\psi }$ of quantum dense coding for three initial states as the function of the damping coefficient *r* with different memory parameters η. (**a**) Bell state, (**b**) Werner state, (**c**) General state.

**Figure 4 entropy-25-00736-f004:**
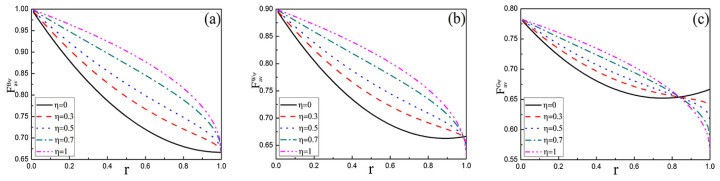
The fidelity $F_{av}^{\psi }$ of quantum teleportation for three initial states as the function of the damping coefficient *r* with different memory parameters η. (**a**) Bell state, (**b**) Werner state, (**c**) General state.

**Figure 5 entropy-25-00736-f005:**
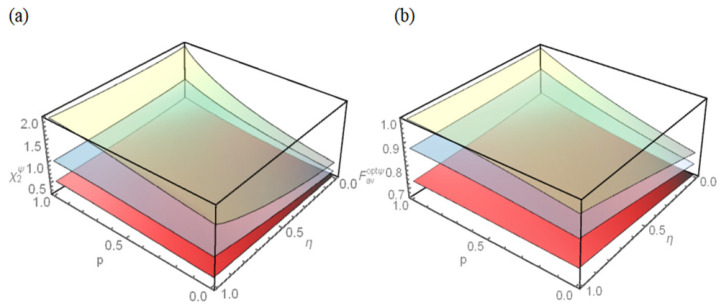
(**a**) The capacity $\chi_{2}^{\psi }$ of quantum dense coding and (**b**) the fidelity $F_{av}^{opt\psi }$ of quantum teleportation for three initial states as the functions of WM strength *p* and the memory parameter η with r=0.5. The upper layer (green) represents the Bell state, the middle layer (gray) represents the Werner state, and the lower layer (red) represents the General state.

**Figure 6 entropy-25-00736-f006:**
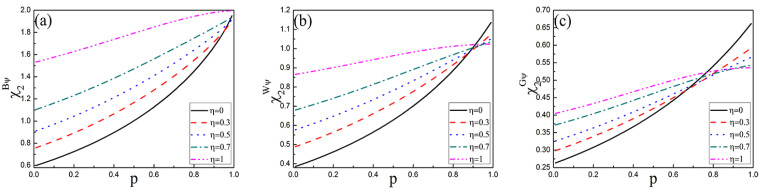
The capacity $\chi_{2}^{\psi }$ of quantum dense coding for three initial states as the function of WM strength *p* with r=0.5 with different memory parameters η. (**a**) Bell state, (**b**) Werner state, (**c**) General state.

**Figure 7 entropy-25-00736-f007:**
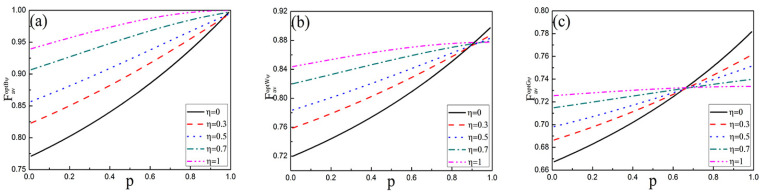
The fidelity $F_{av}^{opt\psi }$ of quantum teleportation for three initial states as the function of WM strength *p* with r=0.5 with different memory parameters η. (**a**) Bell state, (**b**) Werner state, (**c**) General state.

## Data Availability

Not applicable.

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
