# Peer review of "Improving the Capacity of Quantum Dense Coding and the Fidelity of Quantum Teleportation by Weak Measurement and Measurement Reversal"

_entropy, 2023, doi:10.3390/e25050736_

Round 1

Reviewer 1 Report

The authors propose quantum teleportation using weak measurement and measurement reversal. 

Their computation and design of the work may be appropriate, however, there are several unclear parts therefore I would suggest revision as follows. 

1. The abstract of the work should describe the importance of this work more carefully. 

2.  The authors need to clarify which parts of the main work are original and which parts are reviews.

3. For me this is an interesting protocol in the sense that they can teleport a mixed state. The range of applications seems broad, but they are not mentioned at all. The paper would be even better if it emphasized the importance of this protocol and its practical perspective on realistic applications. In this regard, it would be interesting to propose applications to energy teleportation in large-scale networks, which have recently attracted a great deal of attention.   

https://arxiv.org/abs/2301.11884

https://arxiv.org/abs/2301.02666

Author Response

Please see the attachment. The attachment contains the corresponding revision suggestions and the latest revision manuscript.

Reviewer 2 Report

The authors consider two QI protocols, quantum dense coding and
teleportation, under weak measurement / measurement restore (WMR)
protocols, in the presence of an amplitude damping channel "with
memory", with the teleportation scheme assuming perfect EPR
transmission.  Specifically, they calculate the Holevo capacity of
dense coding and the fidelity of quantum teleportation (parameterized
by the channel parameters $r$ and $\eta$) for the three input states
specified explicitly, with different degree of polarization.

In my opinion, the form of AD channel with memory (which actually can
increase entanglement with sufficiently large value of $\eta$) is
highly artificial and non-physical; it is also not clear why perfect
transmission is assumed for the EPR pair.  Further, in my opinion, any
increase of the Holevo capacity or entanglement fidelity as a function
of the "memory" parameter $\eta$ is due to the additional entanglement
created by the channel.  I do not recommend this work for publication.

Author Response

(The authors gave the same response as above.)

Round 2

Reviewer 2 Report

I reviewed authors' responses to the two 1st round reports, as well as the modified manuscript.  In my opinion the work is marginal in quality as the assumptions about the memory in the noise model are artificial and the results are numerical.  It is not clear whether this protocol would give any improvements with a more realistic noise model.  With all that said, I do not oppose a publication of this work (after some optional changes to improve the readability).
